# Isolation and Functional Characterization of a Constitutive Promoter in Upland Cotton (*Gossypium hirsutum* L.)

**DOI:** 10.3390/ijms25031917

**Published:** 2024-02-05

**Authors:** Yang Yang, Xiaorong Li, Chenyu Li, Hui Zhang, Zumuremu Tuerxun, Fengjiao Hui, Juan Li, Zhigang Liu, Guo Chen, Darun Cai, Xunji Chen, Bo Li

**Affiliations:** 1Xinjiang Key Laboratory of Crop Biotechnology, The State Key Laboratory of Genetic Improvement and Germplasm Innovation of Crop Resistance in Arid Desert Regions (Preparation), Institute of Nuclear and Biological Technology, Xinjiang Academy of Agricultural Sciences, Urumqi 830091, China; yangyang@xaas.ac.cn (Y.Y.); lixiaorong@xaas.ac.cn (X.L.); lcy2023123666@163.com (C.L.); 13109911018@163.com (H.Z.); azze128@163.com (Z.T.); lijuan@xaas.ac.cn (J.L.); liuzhigang@xaas.ac.cn (Z.L.); chenguo@xaas.ac.cn (G.C.); cdrhxz104@163.com (D.C.); 2College of Agronomy, Xinjiang Agricultural University, Urumqi 830052, China; 3National Key Laboratory of Crop Genetic Improvement, Huazhong Agricultural University, Wuhan 430070, China; 17806278731@163.com

**Keywords:** *Gossypium hirsutum* L., GUS, constitutive promoter

## Abstract

Multiple cis-acting elements are present in promoter sequences that play critical regulatory roles in gene transcription and expression. In this study, we isolated the cotton *FDH* (Fiddlehead) gene promoter (pGhFDH) using a real-time reverse transcription-PCR (qRT-PCR) expression analysis and performed a cis-acting elements prediction analysis. The plant expression vector pGhFDH::GUS was constructed using the Gateway approach and was used for the genetic transformation of *Arabidopsis* and upland cotton plants to obtain transgenic lines. Histochemical staining and a β-glucuronidase (GUS) activity assay showed that the GUS protein was detected in the roots, stems, leaves, inflorescences, and pods of transgenic *Arabidopsis thaliana* lines. Notably, high GUS activity was observed in different tissues. In the transgenic lines, high GUS activity was detected in different tissues such as leaves, stalks, buds, petals, androecium, endosperm, and fibers, where the pGhFDH-driven GUS expression levels were 3–10-fold higher compared to those under the CaMV 35S promoter at 10–30 days post-anthesis (DPA) during fiber development. The results indicate that pGhFDH can be used as an endogenous constitutive promoter to drive the expression of target genes in various cotton tissues to facilitate functional genomic studies and accelerate cotton molecular breeding.

## 1. Introduction

A promoter is a region of DNA located upstream of the 5’ end of a gene, where RNA polymerase II and transcription factors bind to drive gene transcription and regulate gene expression [1]. Plant gene promoters are classified into constitutive, inducible, and tissue-specific promoters [2]. In plant genetic engineering, constitutive promoters are widely used due to their simplicity and convenience. They are advantageous and desirable in driving the stable expression of target genes across different growth and development stages and tissues and organs in the recipient plants [3].

The most frequently used constitutive promoters are CaMV 35S, isolated from cauliflower mosaic virus [4], the bacterial nopaline synthase (NOS) promoter [5], and the Actin1 promoter of rice actin [6]. Compared with the heterologous expression of target genes, homologous or near-origin promoters are more conducive to increasing the expression intensity of target genes in the genome, resulting in a more stable and reproducible transgene expression [7,8,9]. Endogenous promoters are more advantageous than exogenous promoters in driving the efficient and stable expression of exogenous genes in salt algae [10]. Moreover, transgenic plants with the pKNOX1 endogenous promoter had approximately 2.2-fold higher survival rates at the T3 stage than the p35S promoter [11]. Endogenous promoter and codon optimization resulted in a 6-fold increase in protein expression in moss [12]. The lettuce LsU6-10 promoter could drive the single guide RNA (sgRNA) of a CRISPR/Cas9 system more efficiently than the AtU6-26 promoter, improving gene editing efficiency without any deleterious effects on lettuce plant growth [13]. Therefore, it is crucial to isolate, characterize, and further employ endogenous plant constitutive promoters in transgenic plant research.

Cotton is a globally grown and economically important fiber crop. Cotton fiber is among the main natural resources used by the textile industry, and its fiber quality directly affects the quality of cotton-based textiles [14]. Cotton fibers are unicellular protrusions from the epidermal layer of the ovule. Their differentiation and development can be divided into four stages: initiation, elongation, secondary cell wall thickening, and dehydration and maturation [15]. Cotton fiber formation is a complex process involving the functions and interactions of multiple genes at different stages. Cell-wall-associated transcription factors, as well as expansin, cellulose synthase, sucrose synthase, and actin genes, have been successfully transformed into cotton, resulting in increased production and improved strength and quality of fiber [16,17,18,19,20]. In addition to genes, the roles of promoters in translational research should not be overlooked. The *GhEXPA2* promoter, isolated from Sea Island Cotton 3-79, could drive a strong and tissue-specific GUS gene expression in the fibers during their development, but not in roots, stems, or leaves [21]. The cotton *GhPDF1* gene is highly expressed during fiber initiation and elongation, with the highest transcript levels observed in fibers at 5 days post-anthesis (DPA) [22]. The GhSCFP gene was isolated from a cotton fiber cDNA library, and its promoter transformation with the GUS reporter gene resulted in strong GUS activity in the fibers of transgenic cotton, while no GUS signal was detected in other tissues [23]. Based on a transcriptomics data analysis, the GhACO1 promoter has strong activity and results in high GhACO1 gene expression during fiber elongation in upland cotton [24]. The GhROP6 promoter has very strong activity in cotton fiber, and pGhROP6 could regulate gene expression in fiber and ovule epidermis [25]. Molecular breeding approaches are faster and more efficient in improving cotton fiber quality than traditional breeding. However, only a few cotton endogenous constitutive promoters that could stably and efficiently express downstream target genes have been obtained so far, limiting the efficacy and practical applications of cotton genetic engineering. The use of the exogenous 35S viral promoter has aggravated consumers’ doubts about the safety of genetic modification. Therefore, the investigation and characterization of endogenous constitutive promoters in cotton are important for its genetic improvement and downstream applications. 

*GhFDH* encodes a protein involved in the synthesis of long-chain lipids found in the cuticle. It is a member of the 3-ketoacyl-CoA synthetase (KCS) family, which includes key rate-limiting enzymes in the biosynthetic pathways of very-long-chain fatty acids (VLCFA). Numerous studies have demonstrated that VLCFAs promote cotton fiber elongation [26,27,28,29]. The cotton *FDH* gene is expressed in developing fibers and repressed in hairless mutants, suggesting that it is involved in cotton fiber development [30]. In this study, the *GhFDH* gene promoter was isolated and functionally characterized, revealing that pGhFDH exhibited strong constitutive expression activity. High GUS activity was observed in the roots, stalks, leaves, inflorescences, and pods in the pGhFDH::GUS and CaMV 35S::GUS transgenic *Arabidopsis* plants. Moreover, transgenic cotton plants expressing GUS driven by pGhFDH demonstrated high GUS activity levels in their leaves, stalks, buds, petals, androecium, and endosperm, with enhanced activity, especially during fiber development (10–30 DPA). These findings clearly demonstrate that pGhFDH is a promoter that can drive a strong constitutive expression of downstream genes. Such properties offer valuable application prospects in transgenic cotton breeding and the expression and functional characterization of genes from wild cotton relatives.

## 2. Results

### 2.1. Expression Patterns of GhFDH in Upland Cotton

Based on previously published RNA-seq data [31], a cotton fiber tissue-specific expressed gene *GhFDH* was identified and selected, and its expression profiles in cotton fibers at different developmental periods were examined by qRT-PCR. 

The peak expression of the *GhFDH* gene in fiber cells occurred at 5 DPA and 10 DPA, followed by a rapid decrease after 15 DPA (Figure 1). These observations suggested that *GhFDH* is involved in cotton fiber development, and it is highly expressed, especially during the early stages of fiber development. The *GhFDH* expression levels varied significantly during the different development stages of cotton fiber, with the highest expression levels being observed during the rapid fiber elongation stage (5–10 DPA).

### 2.2. Isolation and Sequence Analysis of the GhFDH Promoter

Specific primers with BP junctions were designed based on the pGhFDH sequence [31]. An 821 bp fragment of the pGhFDH DNA sequence was isolated from the upland cotton cultivar Jin668. The plant expression vectors pGhFDH::GUS and CaMV 35S::GUS were constructed using the GATEWAY [32] method (Figure 2A,B).

The cis-acting elements of the isolated *GhFDH* promoter fragments were identified and annotated using the online software Plant Care [33] (http://bioinformatics.psb.ugent.be/Webtools/plantcare/html/ (accessed on 31 January 2024)) and PLACE [34] (https://www.dna.affrc.go.jp/PLACE/?action=newplace (accessed on 31 January 2024)), finding several cis-elements related to enhancing gene expression in the promoter regions of GhFDH (Figure 2C, Appendix A). Nine TATA boxes and eight CAAT boxes were found in these promoter regions, with the closest TATA box being found at −48 from the start codon of *GhFDH*.

### 2.3. Spatiotemporal Expression Patterns of pGhFDH in Arabidopsis

CaMV 35S::GUS and pGhFDH::GUS were transformed into *Arabidopsis thaliana* by inflorescence infiltration [35], and the pGhFDH-driven GUS expression in different tissues and at different stages of fiber development was observed by histochemical staining of the transformed plants [36]. GUS expression was detected in the roots, stalks, leaves, inflorescences, and pods of the transgenic *Arabidopsis* lines (Figure 3). To assess the promoter activity during plant development, we chose different tissues of two T3-generation homozygous *Arabidopsis* lines for the detection of GUS activity. The results revealed that the pGhFDH::GUS protein had high expression activity in the leaves, stalks, roots, inflorescences, and pods. Notably, the pGhFDH::GUS expression in inflorescences was higher than that of CaMV 35S::GUS (Figure 4). These results suggested that pGhFDH drives the constitutive expression of the GUS gene in *Arabidopsis*. The assay resulted in different lines further confirming the stable and strong expression driven by the *GhFDH* promoter (Figure 4).

### 2.4. Spatiotemporal Expression Patterns of pGhFDH in Transgenic Upland Cotton Plants

To analyze the properties of the pGhFDH promoter driving downstream gene expression in cotton, we transfected the constructed plant expression vectors pGhFDH::GUS and CaMV 35S::GUS into cotton by *Agrobacterium* [37]. After the PCR verification targeting the NPTII, GUS, and promoter sequences (Appendix A), different tissues of transgenic plants were assessed for localized GUS activity. GUS staining was detected in the pGhFDH::GUS and CaMV 35S::GUS cotton leaves, stalks, buds, petals, androgynophore, endosperm, and fibers at different developmental stages (Figure 5 and Figure 6). The pGhFDH-driven GUS was thus expressed across different cotton tissues.

Histochemical analyses were performed on cotton fibers and tissues across different developmental stages to assess the pGhFDH-driven GUS gene activity in different tissues of the transgenic cotton plants. The pGhFDH-driven GUS gene was more highly expressed in all tissues in cotton except for the stalk, bud, and shoot tip, compared to its expression driven by the constitutive promoter CaMV 35S::GUS, with the highest GUS activity levels measured in the anthers and the lowest in the buds (Figure 7, Appendix A). To clarify the patterns of downstream expression regulated by pGhFDH in cotton fibers at different developmental periods, we measured the GUS activity on the day of flowering, 5 DPA, 10 DPA, 20 DPA, 30 DPA, and in mature fibers. The pGhFDH-driven GUS activity was higher than the CaMV 35S-driven GUS activity at all stages. In particular, the highest GUS activity was observed at the fiber elongation stage (10 DPA–30 DPA), which was 6–10-fold higher than that of the CaMV 35S::GUS, followed by a GUS activity decrease at the fiber maturation stage (Figure 7, Appendix A). These results suggest that the pGhFDH promoter is a constitutive promoter with a strong expression-driving ability across all cotton tissues, and has a particularly stronger downstream expression activity than the CaMV 35S promoter in fibers during their development.

## 3. Discussion

Cotton is an economically important fiber crop, and cotton fiber, with its large production and low cost, is the most versatile natural fiber raw material in the textile industry. Cotton fiber quality is mainly controlled by genotype, so it is of great significance to isolate and identify the key genes regulating cotton fiber development. Combined bioinformatics and functional analyses can be adopted to mine the genes that regulate superior fiber quality, which can then be introgressed using transgenic engineering to increase the cotton yield and improve the fiber quality [16,38]. In this study, *GhFDH*, a member of the GhKCSs family, was selected due to its involvement in the regulation of cotton fiber elongation based on transcriptomic data [39]. KCS is the rate-limiting enzyme that catalyzes the condensation reaction in the first step of the VLCFA pathway and determines the final carbon chain length of synthesized VLCFAs [40]. VLCFAs and their derived lipids play important roles in cotton fiber development [29]. The *FDH* subfamily gene *KCS10*/*FDH* is mainly expressed in flowers and young leaves and is involved in VLCFA biosynthesis in epidermal cells [41]. *KCS7, KCS15*, and *KCS19* are characterized by variable expression levels in *Arabidopsis* flowers [42]. We examined the *GhFDH* gene expression levels in cotton fibers at different developmental periods using qRT-PCR, which revealed high expression levels during the 5–10 DPA period. Thus, *GhFDH* is a gene that is highly expressed and active during fiber elongation.

Plant gene expression regulation mainly occurs at the transcriptional level and is regulated by various cis- and trans-acting elements. Promoters are important factors controlling the regulation of gene expression. Plant promoters contain many cis-acting elements that act in coordination with transcription factors to regulate downstream gene expression [2]. Therefore, the comprehensive characterization of the structure and function of plant promoters is conducive to elucidating the regulatory mechanism of gene transcription. It also provides a practical basis for applying genetic engineering approaches to alter the expressions of endogenous or foreign target genes based on the predictive analysis for cis-acting elements present in the 821 bp pGhFDH sequence. The CAAT-box is a common cis-acting element in promoter and enhancer regions that typically exhibits a putative effect in enhancing gene expression. In addition, fifteen DOFCOREZM motifs could be found by PLACE. DOFCOREZM is one of the binding sites of Dof proteins [43,44], and Dof1 and Dof2 have been found to regulate the expression of multiple genes involved in carbon metabolism in maize, such as Dof1 being able to bind to the promoter of both cytosolic orthophosphate kinase (CyPPDK) and a non-photosynthetic PEPC gene to enhance their expression [45]. It has been confirmed that the Dof factor binding sites in subdomain B4 of the CaMV35S promoter are important and contribute to its promoter activity [46]. Although the above cis-elements are only bioinformatically predicted ones, some of them may be truly functional and they may serve as a bases for further experimental characterization and validation [47].

Gene promoters can be classified into three categories based on their spatiotemporal activity: constitutive, tissue-specific, and inducible promoters. In most transgenic plants, constitutive promoters are used to drive the expression of exogenous genes because they are not affected by spatial and temporal constraints and have the advantages of a high efficiency and stability [48]. As a typical constitutive promoter, CaMV35S has been widely used in transgenic crop breeding. CaMV-35S resulted in increased GUS activity in transgenic tobacco [49] and a high expression of downstream genes in transgenic strawberry pollen [50]. Due to its high activity in most tissues throughout the developmental stages of plants, CaMV35S can constitutively induce a high expression of downstream genes, and it is easy to implement by cloning [4,51]. However, certain drawbacks tend to lead to an immune response in host cells in some cases, which can trigger the phenomenon of gene silencing. The PcUbi promoter exhibited higher activity in all tissues of chrysanthemum compared to CaMV 35S [52]. An endogenous constitutive promoter, pOsCon1, identified in rice, was more active than the 35S promoter in the roots, seeds, and callus, and its activity was not affected by the developmental stage or environmental factors [53]. The rice endogenous Ubi promoter resulted in a higher GUS gene expression than the commonly used maize Ubi promoter [54]. Therefore, the availability of endogenous constitutive promoters in different plant species could be crucial to the success of genetic transformation and transgene expression. We fused the target promoter to a GUS reporter gene and determined its expression via GUS activity quantification, with the CaMV 35S::GUS construct as the control. Different tissues and organs (leaves, stalk, buds, petals, anthers, and shoot tips) at six developmental periods (0, 5, 10, 20, 30 DPA, and maturity) were assessed for a comprehensive comparison of the CaMV 35S::GUS and pGhFDH::GUS promoters’ expression activities. Based on the results, pGhFDH drove the constitutive expression of the GUS gene in various *Arabidopsis* tissues (leaves, stems, roots, inflorescence, and pods). A higher expression was observed in the inflorescence under the pGhFDH promoter compared to CaMV 35S. In transgenic cotton, pGhFDH and CaMV 35S resulted in high GUS activity being detected in different tissues at different fiber periods, with pGhFDH resulting in higher GUS activity in all tissues except the stalk, buds, and shoot tips. This indicates that the pGhFDH-driven GUS gene was highly expressed in a stable manner in different tissues at different stages in transgenic *Arabidopsis thaliana* and cotton plants. In this study, the highest expression of the *GhFDH* gene was at 5–10 DPA, but the highest expression of GUS in transgenic material was at 10–20 DPA, and there was some difference between the expressions of *GhFDH* and *GUS*. This might have been due to the fact that the highest expression was at 5–20 DPA during the elongation period of fiber development, and, as an exogenous gene, the expression activity of the GUS gene might have differed at this stage. In addition, this promoter functions mainly in constitutive expression, indicating its potential as a new endogenous constitutive promoter in cotton. Notably, it has been demonstrated that the same constitutive promoter repeatedly driving the expression of multiple exogenous genes may cause gene silencing or the co-repression phenomenon. In this study, we found that the pGhFDH promoter could be a novel option for constructing multi-gene expression vectors, valuable for the application of molecular breeding approaches in improving cotton yield and quality.

## 4. Materials and Methods

### 4.1. Plant Materials

*Gossypium hirsutum* Jin668 and *Arabidopsis thaliana* Col-0 were kindly donated by the National Key Laboratory of Crop Genetic Improvement, Huazhong Agricultural University.

### 4.2. GhFDH Expression Analysis at Different Periods of Cotton Fiber Development

The gene sequence of *GhFDH* (Gh_A13G1665) was retrieved using the Cotton Genome Database (https://yanglab.hzau.edu.cn/CottonMD (accessed on 31 January 2024)). Specific primers were designed according to the gene sequence; the primer sequences are listed in Appendix A.

Tissues from the cotton Jin668 genotype were sampled at the flowering 0 DPA, 5 DPA, 10 DPA, 15 DPA, 20 DPA, and 25 DPA periods. The total RNA was extracted from the samples using the TRIzol reagent (Tiangen, Beijing, China). The nucleic acid concentration was measured using an ultra-micro spectrophotometer (ThermoFisher™ NanoDrop One, Waltham, MA, USA). RNA was reverse transcribed into cDNA using a reverse transcription kit (Tiangen, Beijing, China), and qRT-PCR reactions were performed on a real-time PCR instrument (ABIStepOne™, Foster City, CA, USA). Data were normalized using *GhUbiquitin7* as the endogenous control, and the experiment was conducted in three replicates.

### 4.3. pGhFDH::GUS and CaMV 35S::GUS Expression Vector Construction

Primer-BLAST was used to design primers specific to the *GhFDH* promoter (Appendix A). Cotton leaf genomic DNA was extracted from the cotton cultivar Jin668 using the polysaccharide polyphenol plant genomic DNA extraction kit (DP360) and was used as a template to amplify the fragment of pGhFDH. The plant expression vectors pGhFDH::GUS and CaMV 35S::GUS were constructed using the Gateway cloning technique [32]. The correctly sequenced target fragments of pGhFDH and CaMV 35S were ligated into the pDONORzeo intermediate vector by a BP reaction for transformation. The recombinant plasmids carrying the target fragments were assessed and selected by sequencing. Then, the sequenced target fragments were excised and ligated into the plant expression vector pGWB433 by an LR reaction. Finally, the constructed plant expression vectors pGhFDH::GUS and CaMV 35S::GUS were transformed into *Agrobacterium tumefaciens strain* GV3101.

### 4.4. Genetic Transformation of Arabidopsis thaliana Plants

*Arabidopsis thaliana* was transformed by the *Agrobacterium*-mediated floral-dip transformation method [33]. Wild-type *Arabidopsis thaliana* (Col-0) seeds were planted in an illuminated culture chamber (22 °C, 14 h light/10 h dark, 50 μmol/m^2^·s photon flux density of photosynthetically active radiation light intensity) and transformed with both pGhFDH::GUS and CaMV 35S::GUS expression vectors. *Arabidopsis* inflorescences were inoculated by resuspending the activated *Agrobacterium* GV3101 carrying the target vector with infiltration buffer (5% sucrose, 0.01% Silwet L-77). After inoculation, the plants were left in dark, moist conditions for one day. The inoculation of the nascent inflorescences was repeated once a week later to increase the number of transformants, and the seeds were collected when they became ripe. The collected T0 generation seeds were sterilized, placed on 1/2 MS medium containing kanamycin (Kan), and cultured in an illuminated culture room. After two weeks, green plants, resistant to Kan, were selected and transplanted into culture soil (containing vermiculite: nutrient soil: flower soil = 1:1:2). Following this process, T3 generation plants were generated and used for GUS staining and detection.

### 4.5. Genetic Transformation of Cotton

*Agrobacterium*-mediated genetic transformation of hypocotyls [35] was used to transform the upland cotton genotype Jin668. The seed coat of the cotton seeds was peeled off. The seeds were then sterilized with a 10% NaClO solution for 15 min and subsequently rinsed with sterile water until there was no foam. Then, the sterilized seeds were germinated in a culture medium. After one week, the hypocotyls were cut and used as explant material. *Agrobacterium* containing the recombinant plasmid was centrifuged and resuspended in the Mannitol Glutamate Luria (MGL) solution. Acetosyringone was added at a ratio of 4000:3, shaken at 180 rpm at 28 °C for 20 min, and then removed for explant infection. The hypocotyls were impregnated for 10 min and then transferred to a co-culture medium and incubated at 24 °C for 3 days in the dark. Then, the hypocotyls were transferred to a Murashige and Skoog Provitamin B_5_ (MSB) screening medium, which was replaced every 20 days. The formed embryonic calluses were then transferred to a somatic embryo maturation medium for incubation. After the mature somatic embryos were grown, the seedlings were transferred to a seedling growth medium and then to the greenhouse for cultivation when the seedlings reached 3~4 cm in height. They were grown in sterile pots (containing vermiculite:nutrient soil:flower soil = 1:1:2) in a plant culture room at a 25 °C/21 °C day/night temperature under a 75 μmol/m^2^·s photon flux density of photosynthetically active radiation, a relative humidity of 60–75%, and a 16 h day/8 h dark photoperiod, with regular watering. T3 generation plant tissues were collected and used for GUS staining.

### 4.6. Determination of GUS Activity and Histochemical Staining

Different tissues and fibers from different periods of transgenic plants were placed into the GUS staining solution for overnight staining at 37 °C and stored in formaldehyde-alcohol-acetic acid (FAA) fixative. After decolorization in 70% alcohol, the GUS gene expression was observed under an optical microscope Leika MZFLIII (Hesse, Germany), and pictures were taken and stored. Each experiment consisted of at least 50 *Arabidopsis* plants and 10 cotton plants in three replicates.

The method reported by Jefferson [34] was used to quantify the GUS activity. The GUS enzyme activity was measured immediately in a 1 mL centrifuge tube by adding 400 µL of GUS protein extraction buffer, 100 µg of GUS protein, and 10 µL of 40 mmol L^−1^ 4-MUG. The reaction was terminated by adding 1.6 mL of reaction termination buffer at 37 °C for 1 h. The GUS enzyme activity was measured immediately in a 1 mL centrifuge tube. The sample absorbance was determined at 365 nm excitation light and 455 nm emission light with 50 nmol L^−1^ 4-MU as the standard. At least three biological replicates were performed for each sample. The GUS activity was expressed as pmol 4-MU μg^−1^ min^−1^.

## 5. Conclusions

The pGhFDH promoter is a constitutive promoter with a strong capacity to drive downstream gene expression in all tissues in cotton. Notably, it exhibited higher activity in cotton fibers compared to the CaMV 35S promoter during their development. Our results suggest that pGhFDH has great potential to be implemented in crop molecular breeding for constitutive gene expression and to avoid gene silencing caused by the use of exogenous promoters.

## Figures and Tables

**Figure 1 ijms-25-01917-f001:**
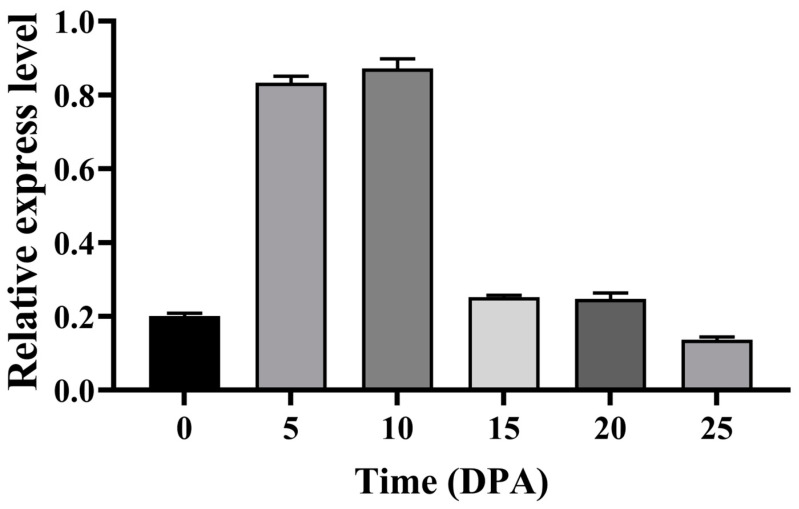
Expression of the *GhFDH* gene at different time points during cotton fiber development. Three technical and biological repeats were performed for each time point. The cotton endogenous gene *GhUb7* was used as a reference standard.

**Figure 2 ijms-25-01917-f002:**
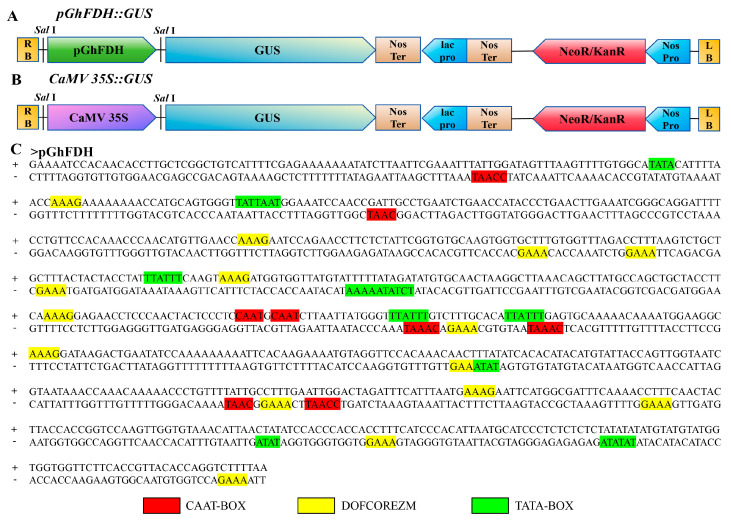
Schematic illustrations of the expression vector constructs and the putative cis-acting elements of the *GhFDH* promoter. (**A**): Schematic illustration of the pGhFDH::GUS expression vector; (**B**): schematic illustration of the CaMV 35S::GUS expression vector; and (**C**): the location of putative cis-acting elements in the *GhFDH* promoter predicted by the PlantCARE and PLACE database.

**Figure 3 ijms-25-01917-f003:**
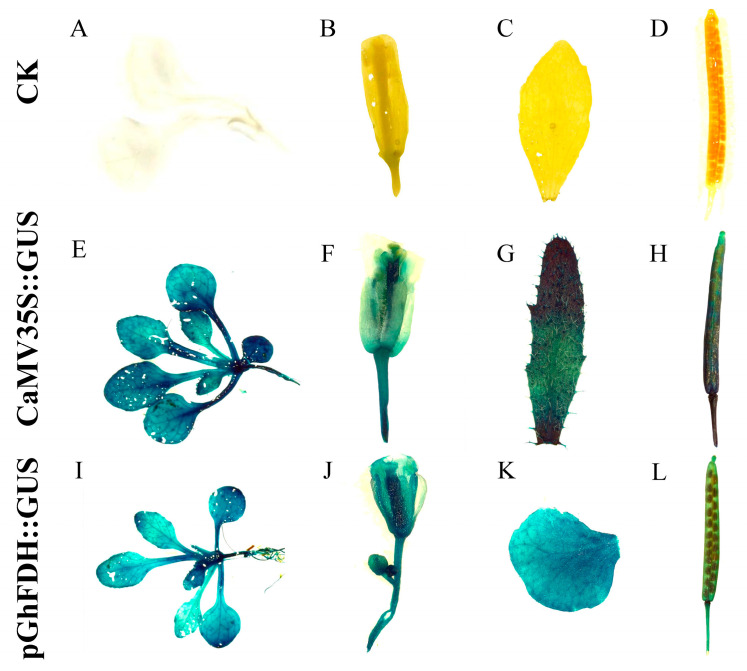
β-glucuronidase (GUS) staining of transgenic *Arabidopsis thaliana* tissues. (**A**–**D**): The seedlings, flowers, leaves, and seeds of wild-type *Arabidopsis* plants; (**E**–**H**): the seedlings, flowers, leaves, and seeds of CaMV35S::GUS T3-generation transgenic *Arabidopsis* lines; and (**I**–**L**): the seedlings, flowers, leaves, and seeds of pGhFDH::GUS T3-generation transgenic *Arabidopsis* lines.

**Figure 4 ijms-25-01917-f004:**
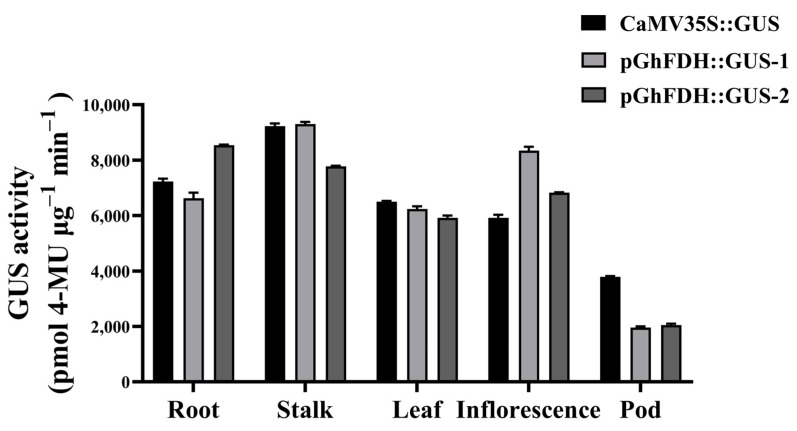
β-glucuronidase (GUS) activity in different tissues of CaMV 35S::GUS and pGhFDH::GUS transgenic *Arabidopsis* lines. GUS activity was expressed as pmol 4-MU per μg protein. 4-MU, 4-methylumbelliferone.

**Figure 5 ijms-25-01917-f005:**
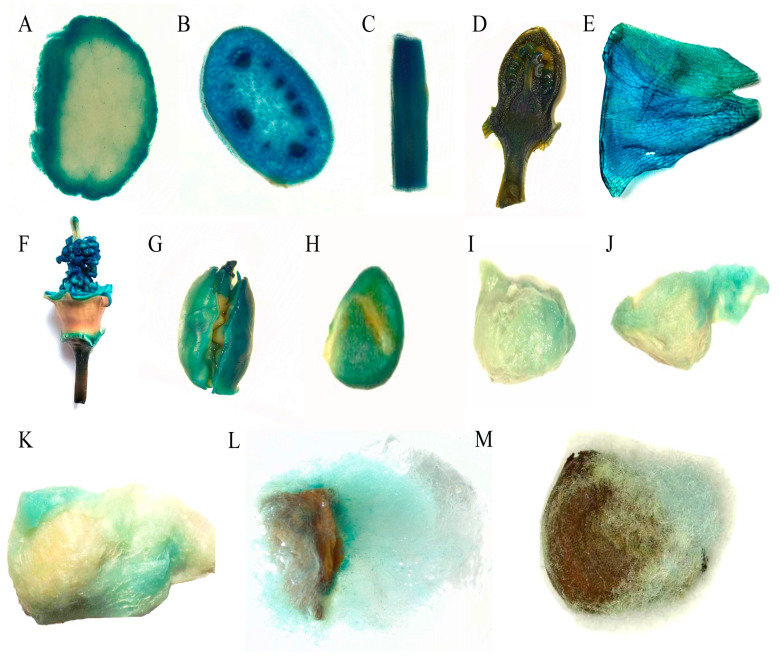
β-glucuronidase (GUS) staining of transgenic cotton plants expressing the CaMV35S::GUS gene. (**A**–**G**): Leaf blades, stem longitudinal sections, stem transverse sections, bud longitudinal sections, petals, androgynophore, and embryo of the CaMV35S::GUS-expressing cotton plants; (**H**–**M**): 0 days post-anthesis (DPA), 5 DPA, 10 DPA, 20 DPA, 30 DPA, and mature fibers of the CaMV35S::GUS-expressing cotton plants.

**Figure 6 ijms-25-01917-f006:**
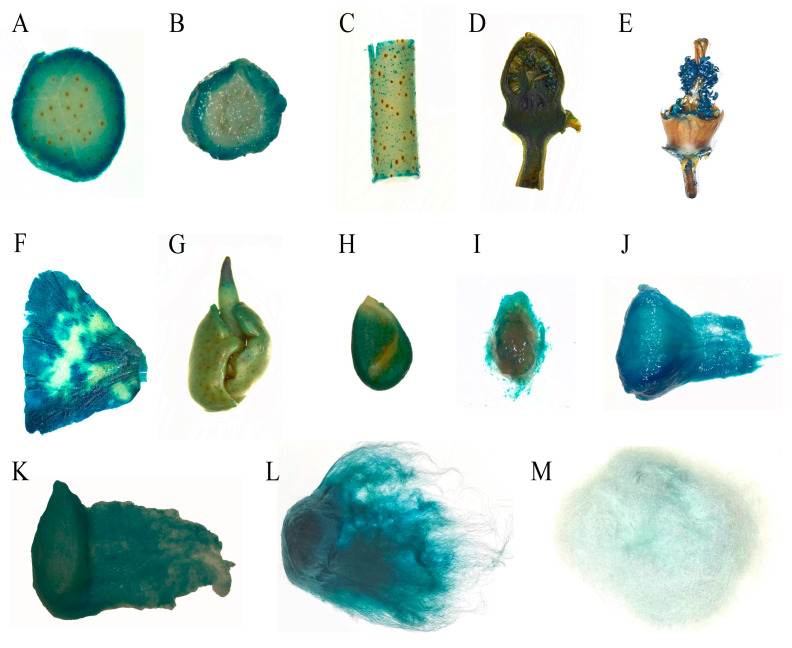
β-glucuronidase (GUS) staining of transgenic cotton plants expressing the pGhFDH::GUS gene. (**A**–**G**): Leaf blades, stem longitudinal sections, stem transverse sections, bud longitudinal sections, petals, androgynophore, and embryo of the pGhFDH::GUS-expressing cotton plants; (**H**–**M**): 0 days post-anthesis (DPA), 5 DPA, 10 DPA, 20 DPA, 30 DPA, and mature fibers of the pGhFDH::GUS-expressing cotton plants.

**Figure 7 ijms-25-01917-f007:**
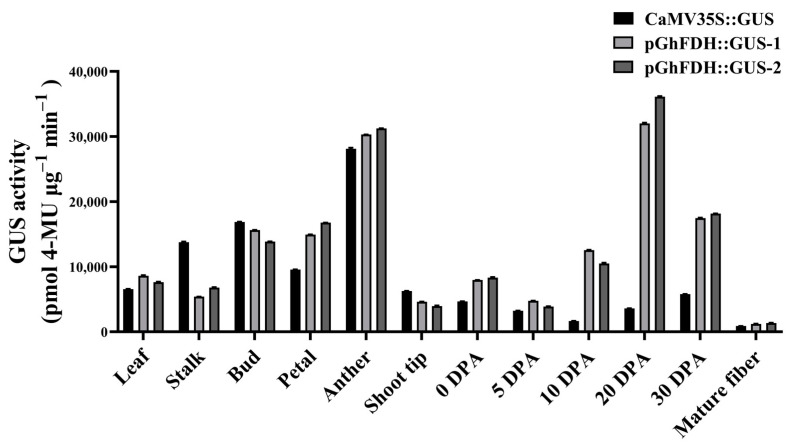
β-glucuronidase (GUS) activity in different tissues and different fiber developmental stages of CaMV 35S::GUS and pGhFDH::GUS transgenic cotton lines. GUS activity was expressed as pmol 4-MU per μg protein. 4-MU, 4-methylumbelliferone. DPA: Days post anthesis.

## Data Availability

The datasets presented in this study can be found in Appendix A.

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
