# Peer review of "Isolation and Functional Characterization of a Constitutive Promoter in Upland Cotton (Gossypium hirsutum L.)"

_ijms, 2024, doi:10.3390/ijms25031917_

Round 1
Reviewer 1 Report
Comments and Suggestions for Authors
Authors isolated a new promoter sequence which drives expression across several tissues. The expression is comparable to the well-known CaMV35S constitutive promoter. The new promoter could be of use, assuming its high expression at 10 to 20 days, as expected from the original expression pattern of the gene.
Still, here I detected several inconsistencies:
- author's isolated promoter of 821 nucleotides but showing on the figure 2 promoter over thousand BP length.
-all sequences of promoters etc must be presented.
- Original promoter GhFDH gene has a peak expression at 5 to 10 days but it's cloned version has a peak from 10 to 20 days -to discuss.
- cis-module analysis: All your predictions have a high rate of false positives and most of your predicted motifs, if not all, are actually false. Since you are not using this motif analysis in your experiment this part should be removed at least to the supplementary or at best at all. You may present it in the next work where you will check the functionality of this motifs. See this as example how it can be done (PMID: 27171245 )
- Notably, pGhFDH::GUS expression in inflorescences was significantly higher than that of CaMV 35S::GUS (Figure 4). - must be removed as it's not true! - expression is very similar!
- Discussion: see 1st paragraph!
Reviewer 2 Report
Comments and Suggestions for Authors
The reviewed manuscript, titled "Isolation and functional characterization of a constitutive promoter in upland cotton(Gossypium hirsutum L.)” by Yang et al. submitted to IJMS is a brief study that is focused on a description of a promoter from the commercially important cotton plant. Overall, the only major critique of this work is that it is incredibly brief and does not cross the threshold necessary to be classified as an article or report. I would characterize this as a brief communication at best.
Major comments:
There is no functional dissection of this in the slightest. Setting up a series of genetics experiments, such as a linker scanning assay to verify the predictions regarding the computational identification of each of the cis acting elements presented in figure 2C would be incredible beneficial and useful. This would allow for a greater functional characterization of this promoter (as is the title of the submitted work). As submitted in this form I am surprised that the paper was not desk rejected - it does not comprise a completed study that is suitable for publication in this present form
Minor edits:
Line 16: introduce acronyms in their first instance – e.g. what is the FDH gene? Do this throughout the work.
Line 199-202: These are instructions from the template supplied to the authors. Please remove work that is not part of your article.
Comments on the Quality of English LanguageEnglish needs some revisions.
Round 2
Reviewer 1 Report
Comments and Suggestions for Authors
Promoter:
First I was positive about the paper but after the revision I have a feeling that the authors want to hide some details. Therefore I would ask for really detailed data on experiments in major revision.
Comments 2: Authors present sequences as image and this concerns me very much, because this way readers cannot copy the sequence to check it.
Comments 4: cis-module analysis: This is very suspicious: “ we used a different prediction site for the promoter-specific elements of the promoter for this prediction and the results were close.“ - You must present these results! In every detail !
Actually what you are doing is contradicting your statement that your promoter is constitutive. if your promoter has these regulator elements it cannot be constitutive it will be tissue specific.
Comments 5: pGhFDH::GUS-2 is comparable to 35S, fluctuation between 1 and 2 is big, as in root and stalk. What are error bars in that case, please give data how it is calculated.
Here, I agree with the first reviewer that there are no details given which help readers follow and fully understand what exactly was done.
Reviewer 2 Report
Comments and Suggestions for Authors
The authors have made several improvements to the work as outlined by the reviewers. Overall the quality and the clarity of the English is substantially improved by the revisions and edits that they had made. The only major concern that i have is the rather superficial identification on cis regulatory promoter motifs, specifically figure 2C. The typical follow up when attempting to identify and characterize promoter elements is to perform any type of funtional assay to validate that the predictions reveal bona fide elements, this component of the work is very weak. There are many compounding factors that are layered on top of nucleic acid elements, including epigenetics such as nicleosome positioning, histone modification etc. As such, it is necessary for the authors to provide more information and statistical analysis to allow the readers as much information as possible when considering this portion of the work. Specifically missing are the consensus motifs used for prediction and the p/e values for each motif. This would allow for at least a cursory comparison of whether each motif is likely to actually play a role in regulation based on its identify and information content.
With that i dont have any additional reservations about publication, however the work is very low impact, simply identifying an expression system in a rather superficial manner.
Round 3
Reviewer 1 Report
Comments and Suggestions for Authors My concern was to emphesize the difference between bioinformatic predictions and reality. In principle the paper is ok. Last thing — to better explain the difference between computationally predicted motifs and identification of real motifs which always includes wet-lab experiments, in the discussion (lines 256-265) should be included somthing like this: Although, the above cis-elements are only bioinformatically predicted ones, some of them may be truly functional and they may serve as a bases for further experimenetal characterisation and validation [PMCID: PMC3465240]. I will leave authors to deside wether to include this.Author Response
Please see the attachment

Round 4
Reviewer 1 Report
Comments and Suggestions for Authors
Authors made all required changes.